# Differences in Clinical Tests for Assessing Lateral Epicondylitis Elbow in Adults Concerning Their Physical Activity Level: Test Reliability, Accuracy of Ultrasound Imaging, and Relationship with Energy Expenditure

**DOI:** 10.3390/ijerph20031794

**Published:** 2023-01-18

**Authors:** Marcos Martins Soares, Patrícia Colombo Souza, Ana Paula Ribeiro

**Affiliations:** 1Health Science Post-Graduate Department, School of Medicine, University Santo Amaro, São Paulo 04829-300, Brazil; 2Medicine and Physical Therapy Department, School of Medicine, University of São Paulo, São Paulo 05360-160, Brazil

**Keywords:** lateral epicondylitis, physical activity, pain

## Abstract

Background: Physical activity (PA) can generate physical stress on the musculoskeletal system. Thus, the aim of the current study was to assess the influence of the level of physical activity on clinical tests used in the diagnosis of lateral epicondylitis in adults, intertest reliability and accuracy based on ultrasound imaging, and relationship with energy expenditure. Methods: 102 adults with lateral epicondylitis were assessed via an International Physical Activity Questionnaire and divided according to PA level: low (*n* = 19) moderate (*n* = 42) or high (*n* = 41). Pain (visual analog scale), Cozen’s and Mill’s clinical tests and ultrasound exams were performed. Results: The Cozen’s and Mill’s tests differed among PA levels. Excellent reliability was found for Mill’s test for PA levels and the ultrasound exam (low ICC = 1.0, moderate ICC = 0.82 and high ICC = 0.99). Good reliability was found for Cozen’s test (low ICC = 0.80, moderate ICC = 0.74 and high ICC = 0.73), but with significant differences between the ultrasound exams. The Cozen’s and Mill’s clinical tests had no relationship to the level of energy expenditure for PA levels. Conclusion: Mill’s test was reliable and accurate for the PA levels. Intertest reliability was poor for the PA levels. Mill’s test proved to be accurate based on the ultrasound exam. The pain caused by the tests was not related to the level of energy expenditure.

## 1. Introduction

Currently, lateral epicondylitis is one of the common musculoskeletal disorders of the segmental region of the elbow, affecting from 1% to 3% of the adult population of both sexes, with 75% of cases in the age group from 35 to 55 years [1]. The incidence of the pathology of epicondylitis is seven times more likely to occur in the lateral than in the medial epicondyle, the former being the elbow joint dysfunction most recurrent in orthopedic clinics, resulting from intense and repetitive physical efforts in labor or sports practices [2].

Scientific evidence shows that lateral epicondylitis mainly affects two distinct groups of adults: the first, constituting 5% of those affected, are athletes who overload the elbow extensor muscle activity, usually with specific sports practices, such as tennis, golf and squash [3]. The second group, which constitutes 95% of the affected individuals between the ages of 35 and 55, present symptoms with an insidious and progressive onset, largely generated by repetitive work activities associated with physical labor [4]. Despite being distinct, the second group presents difficulties for the clinical diagnosis of epicondylitis, given the various types of repetitive efforts associated with the work and physical activity.

Epicondylitis has been defined as a musculoskeletal dysfunction of inflammatory and degenerative origin that affects the extensor tendons of the wrist and fingers, often known as “tennis elbow” [5]. Many of the functional impairments resulting from lateral epicondylitis are due to repetitive efforts by the wrist and finger extensor muscles, which are dependent on the level of demand of the physical practice performed by the patient [6]. To understand the level of physical effort in the patient’s work, this study considers the labor practice (at home or work with repetitive effort), the excess load imposed by sports training and the continuous demand for physical activity, which can individually or together increase the intensity of the demand for repetitive elbow effort, especially in the extensor musculature, leading to the development of inflammatory conditions in the tendon or aggravation of clinical conditions already present, with recurrences of the disease [7].

The kinematics of the wrist and electromyography in patients with lateral epicondylitis have been shown to result in an increase in muscle recruitment of the wrist extensor muscles, and consequently in the demand for resistant eccentric muscle strength [8]. The intensity of the muscular force generated by the exercise or increases in imposed physical activity will reproduce the demand for stretching force on the tendon of the extensor musculature, characterizing one of the mechanisms of greatest impact for the onset of the disease [9].

The incidence rate and recurrence of the disease increase with age, female gender, duration of sports practice and frequency of weekly training [9,10,11]. Work activities such as scanning for prolonged periods, cutting objects, answering phones, painting and overexertion at work or physical activity, are reportedly associated with increased tension on the extensor tendon in the elbow region [12,13,14]. Thus, recent research has highlighted the importance of accurate and early diagnosis for a better clinical picture of lateral epicondylitis. In most cases, the diagnosis of lateral epicondylitis can be assigned clinically, with subsequent conservative treatment already in primary health care. However, this early diagnosis is not routinely assigned, nor is it widely clarified through specific imaging exams, which prolongs pragmatic resolution and enables the disease to evolve from an acute to chronic clinical conditions with tendon degeneration [14,15].

To improve the chances of early diagnosis, physical examination of patients with lateral epicondylitis is paramount, given that local sensitivity in the extensor muscle–tendon structures in the lateral epicondyle is the point of the greatest pain symptoms reported by patients [16]. Among the specific clinical tests for the diagnosis of lateral epicondylitis, the Cozen’s test and the Mill’s test stand out; these are pain-causing maneuvers in the origin of the extensor tendons, located in the lateral epicondyle [17,18,19]. These clinical tests are the most widely used in daily clinical practice [19,20,21]. However, few studies report their diagnostic accuracy in relation to imaging, and no studies have reported the influence on the test results of the level of effort in the exercise practiced by the patient [22]. A study has previously attracted attention due to the inference that the Cozen’s test may be excellent in ruling out the diagnosis of lateral epicondylitis, whereas Mill’s test effectively confirms the diagnosis [14,22].

Among the imaging tests available, ultrasound is considered the first-line diagnostic test for confirmation of lateral epicondylitis [14]. Ultrasound is an accessible and radiation-free examination, and despite being operator-dependent, it can confirm the clinical suspicion of lateral epicondylitis and exclude other causes of pain in the epicondyle in addition to providing information on the extent and severity of the condition [22,23]. Changes in thickness and echogenicity, calcifications and tears can be detected in lateral epicondylitis by ultrasound [14,15]. Miller et al. [24] reported a sensitivity of 64–82% and a specificity of 67–100% of ultrasound exam for the diagnosis of lateral epicondylitis.

Most studies that report the association of lateral epicondylitis with physical risk factors at work do not consider physical activities, hobbies, and sports activities performed outside of work [25]. Therefore, there is still a lack of understanding as to whether this extra physical stress can cause any interference in the diagnostic evaluation of these patients affected by the disease and who already have an overload due to work activities and/or physical activity. Thus, the primary purpose of the current study was to verify the influence of physical activity level on lateral epicondylitis using clinical diagnostic tests (provocation test: Cozen and Mill). The following secondary purposes were defined: the inter-diagnostic test reliability and its validity were verified using the ultrasound exam as the gold standard, different levels of physical activity, the correlation analysis of the pain provoked by the clinical diagnostic tests (Cozen and Mil), and the energy expenditure obtained for each PA level of the patients. Our hypotheses were: (a) the level of physical activity interferes with diagnostic tests for lateral epicondylitis; (b) the level of physical activity influences the inter-clinical test reliability (Cozen and Mill) for the diagnosis of the disease and its accuracy by ultrasound examination of the musculoskeletal tissue; (c) the energy expenditure obtained for each level of physical activity is related to the pain induced by clinical tests (Cozen and Mill).

## 2. Materials and Methods

This study has a cross-sectional design. The sample consisted of 102 adults seen at an orthopedic clinic in the southern region of São Paulo. All participants signed a free and informed consent form, agreeing to undergo the evaluations of this research, previously approved by the local Ethics and Research Committee (number: 3367645).

The characteristics of the 102 adults evaluated were: mean age of 45.7 ± 7.6 years, height of 1.6 ± 0.8 m, and body mass of 74.3 ± 13.8 kg; 67 (65.6%) were females and 35 (34.5%) were males. In total, 79 adults (77.5%) performed office work, including administrative assistant, digital computer technician, finance, marketing technician, and telephone assistance, and 23 adults (22.5%) were teachers, health professionals (physician, nursing, and physical educator), and occupied management positions. The sports activities practiced by the participants were walking, swimming, bodybuilding, running, bicycling, dancing, aerobics and muscle resistance training with Pilates.

The eligibility criteria for participation in this study were the following: adults aged between 18 and 55 years, presence of moderate to high intensity pain in the elbow, practicing physical activity and agreeing to participate in the research. Exclusion criteria were the following: drug infiltrations in the lateral epicondyle region, presenting vestibulocochlear diseases, uncontrolled cardiac and/or respiratory arrhythmias, convulsive and neurological syndrome, as well as musculoskeletal disorders, such as diabetic neuropathy, osteoarthritis, rheumatoid arthritis and tissue lesions (cutaneous ulcers of any etiology) that are functionally limiting. Additionally, the use of prostheses and/or orthoses in upper limbs or fractures in the last 6 months was also considered an exclusion criterion, that is, not maintaining a good general health condition; this ensured the interpretations of the assessments was not biased.

Participants were divided into three groups according to their PA level. Group 1 was composed of 19 participants with a low PA level; group 2 was composed of 42 participants with a moderate PA level; and group 3 was composed of 41 participants with a high PA level. The level of PA was measured by using the International Physical Activity questionnaire (IPAQ). The IPAQ facilitates estimation of time spent walking, engaging in moderate and vigorous intensity PA, or sitting, both during the week and on weekends. The questionnaire covers multiple domains: work, travel, housekeeping and leisure in one typical week or the previous seven days. Detailed information on duration (minutes/day) and frequency (days/week) was collected for the different dimensions of physical and sedentary activity in all domains. Activities performed for at least ten continuous minutes in the previous week were considered. Intensity, expressed in terms of working metabolic equivalents (METs), was determined in accordance with the guidelines for data processing and analysis that form part of the IPAQ. To calculate PA scores in the different domains mentioned above, the following formula was used: Intensity (METs) *Duration (minutes/day) * Frequency (days/week). The total PA score was generated as the sum of the scores (work + travel + domestic + leisure) in METs/minute/week and the final classification of PA levels was obtained using the SAS statistical software, version 9.1 (SAS, Raleigh, NC, USA) [26].

**Evaluation of pain:** The symptom of low back pain was assessed using the visual analogue scale (VAS); the scale ranges from 0 to 100 mm, with 0 indicating absence of pain and 100 indicating unbearable pain [27].

**Evaluation of clinical tests:** Cozen‘s and Mill’s tests, specific clinical tests for lateral epicondylitis, aim to reproduce the pain experienced by the patient. To perform the Cozen’s clinical test, a participant was asked by the evaluating physician to perform an active wrist extension against resistance. The test was performed with the participant sitting in a chair without armrests, with the shoulder slightly adducted and the elbow in 90° of flexion, with the forearm in pronation and the wrist in a neutral position. Participants were instructed to perform this maneuver with as much muscle effort as possible to the limit of bearable pain, twice, with an interval of two minutes between each attempt. The test was positive if the patient reported pain in the lateral epicondyle, origin of the extensor muscles of the wrist and fingers [28,29,30,31]. To perform the Mill’s clinical test, the participant was asked to perform passive wrist flexion with their hand closed against resistance to the evaluator’s movement. The test was performed with the participant sitting in a chair without armrests, with the shoulder slightly adducted and the elbow in extension, with the forearm pronated and the wrist flexed. Participants were instructed to perform the maneuver with as much muscle effort as possible to the limit of bearable pain, twice, with an interval of two minutes between each attempt. The positivity of the test was the presence of moderate to high intensity pain in the lateral epicondyle [29,30,31].

**Evaluation of a diagnostic ultrasound examination for epicondylitis:** An elbow joint ultrasound, specifically of the extensor tendon, was performed using an ultrasound device. During the examination, the patient remained seated with their elbow resting on the stretcher. Each ultrasound examination was previously requested by the orthopedist specialized in elbow, wrist, and hand, and performed in an image examination laboratory with the specific protocol using a dedicated ultrasound device. A specific algorithm was used to reduce spicules and disorganization due to image discontinuity, as well as to improve the contrast resolution, the distinction of the borders, and the margin interfaces of the extensor tendon. For better standardization, high frequency linear transducers were used, ideal for evaluating surface structures with an ultrasound frequency between 5 and 12 MHz. The analysis of the images of the ultrasound examination was performed by experienced radiologists, with the titles of specialist and specific training in examinations of the musculoskeletal system [32,33,34,35].

**Inter-test reliability analysis:** The assessment of inter-clinical reliability for different groups of physical activity was performed using the intra-class agreement index (ICC) type 1.1 [36,37,38]. The two tests were conducted in random order, respecting a time interval of 30 min between one test application and the next. ICC values less than 0.40 indicated worse reliability, values between 0.40 and 0.59 indicated poor reliability, those between 0.60 to 0.75 indicated good reliability, and those equal to or greater than 0.76 indicated excellent reliability. The standard error of measurement (SEM) and the 95% confidence interval were calculated.

**Validity analysis between clinical tests and ultrasound examination:** The evaluation of concurrent validity of the clinical tests with each positive and negative ultrasound image examination was performed using Pearson’s correlation test followed by the Kappa agreement index to determine the validity of the clinical tests (Cozen’s and Mill’s). Ultrasound is considered as one of the gold standard exams to detect inflammatory changes in of the tendon, especially in cases of lateral elbow epicondylitis. Kappa values between 0.41 and 0.60 were considered to indicate weak agreement, those between 0.61 and 0.80 indicated good agreement, and those between 0.81 and 1.0 indicated excellent agreement. Bland–Altman analyses were performed to determine the limits of agreement between the clinical tests and the ultrasound images.

**Statistical analysis:** The sample size calculations to determine that the sample should comprise 110 adults were based on an equation for the correlation coefficient (between the ultrasound exam and MET, obtained for studying PA). A moderate effect size (F = 0.25), 80% power and a 5% significance level were used in the calculation. However, the final sample included only 102 participants, since eight adults who had previously been evaluated did not attend on the day and time scheduled for the ultrasound exam.

All statistical analyses were performed using SPSS Statistics version 24 (IBM, Chicago, IL, USA). The normality of the data was verified using the Shapiro–Wilk test. To compare the anthropometric means between the groups of physical activity, one-way analysis of variance (ANOVA) was used, and to evaluate the differences between the clinical tests measured for each group of physical activity level, the paired t-test for dependent measures was used.

The inter-test reliability between the physical activity groups was performed using the intra-class agreement index (ICC), followed by the analysis of the standard error of the measurement (SEM) determined by the formula SD × √1 − ICC, as well as the 95% confidence interval for each ICC. To verify the validity of the clinical tests in relation to the ultrasound exam, the agreement index was used in the Kappa test and Bland–Altman analyses [39].

Simple linear regression is a mathematical technique used to model the relationship between a single independent predictor variable and a single dependent outcome variable. Thus, in the current study, the simple linear regression analysis was used to predict the dependent variables (pain provoked by the clinical diagnostic tests: Cozen’s and Mill’s) and the level of physical activity (energy expenditure score measured by IPAQ). The dependent variable (pain Cozen’s and pain Mill’s) was entered in the model by correlation coefficients that were higher than 0.30. For all of the analyses, we adopted *p* < 0.05.

## 3. Results

The anthropometric variables of age, height, body mass, body mass index, and gender together with the clinical variable of disease duration did not show statistically significant differences among groups of subjects in the study. Only the time of physical activity practice was significantly different among the groups—the high PA group had a much longer practice time when compared with the low and moderate PA level groups (Table 1), showing that the effort to practice between groups was different.

Table 2 shows that the results of the Cozen’s and Mill’s clinical tests differed among the groups of physical activity levels. Mill’s test reported greater elbow pain with lateral epicondylitis, regardless of the adult’s effort in the practice of physical activity. This finding could be of great importance in the effective diagnosis of the disease, especially during the clinical routine evaluation of the patient.

Table 3 shows worse inter-test reliability in the low-PA group, whereas the moderate and high-PA groups exhibited poor inter-test reliability in the diagnosis of lateral elbow epicondylitis, regardless of the PA practice efforts. The tests did not show good accuracy of agreement between them.

According to Table 4, Mill’s test showed excellent agreement of validity for the different levels of PA: low (Kappa = 1.0), moderate (Kappa = 0.82) and high (Kappa = 0.99), with possible positive diagnosis of lateral epicondylitis, evaluated by the image of the elbow tendon using ultrasound. Notably, there was no significant difference between the PA groups when comparing the positive and negative ultrasound results, with a low difference of agreement expressed by the Bland–Altman analysis (ranging between 0.0; 0.11 and 0.02, respectively). Cozen’s test, however, demonstrated good agreement with the result of the ultrasound exam in the different PA groups: low (Kappa = 0.80), moderate (Kappa = 0.74) and high (Kappa = 0.73), and showed significant differences between positive and negative ultrasound exams. This is further supported by the high difference in agreement obtained with the Bland–Altman analysis (0.14–0.17) (Table 4).

Multiple linear regression analysis showed that the pain symptom caused by both the Cozen’s and Mill’s tests has no relation with the level of energy expenditure (MET/min/s) of the different levels of PA practice. This suggests that the pain associated with an inflammatory response of the tendon with lateral elbow epicondylitis did not change in intensity according to the individual level of physical fitness of the adult. Therefore, PA levels do not alter the pain caused by clinical tests during the diagnosis of the disease (Table 5).

## 4. Discussion

The purpose of the present study was to verify the influence of physical activity on the inter-clinical reliability for diagnosis of lateral epicondylitis and the inter-test validity with the ultrasound examination of the affected adults. The relationship between the energy expended (MET/min/week) and the symptom of pain on the tendon caused by the clinical tests of Cozen’s and Mill’s clinical tests for the diagnosis of lateral epicondylitis was also verified. The main results showed the low inter-test reliability of the Cozen’s and Mill’s tests at the different levels of PA (low, moderate, and high). Mill’s test more accurately reflected the results of the ultrasound examination of the affected tendon and was more effective in provoking pain symptoms compared to Cozen’s test in all the groups. However, the intensity of the pain symptoms caused by the clinical tests was not associated with the level of energy expenditure (MET/min/s) for the different levels of PA practice.

The pain involved in the clinical framework of lateral epicondylitis is one of the primary symptoms in the diagnosis of the pathology [28]. Thus, in a systematic review that assessed the accuracy of clinical tests for elbow pathologies, including lateral epicondylitis, concluded that the greatest efficiency of the diagnosis of a pathology is obtained with a combination of clinical tests [19]. Despite verifying the importance of using more than one clinical test for the physical examination of a patient with lateral elbow epicondylitis, in the current case, we can verify an important finding that Cozen’s and Mill’s clinical tests do not differ in results between different groups of patients with different levels of physical activity, in addition to showing a low inter-test agreement in the symptomatology of the muscle–tendon structures of the lateral epicondyle. According to some authors, the anatomical region of the tendon structures of the lateral epicondyle is the most effective pain-inducing site for diagnosing lateral epicondylitis [16,17,18,19]. Based on this rationale, the results of this study showed that the Mill’s test was more expressive than Cozen’s test in promoting a positive diagnosis of the lateral epicondyle pain symptoms. This corroborates the findings observed by Saroja et al. [22], who described the Mill’s test as an effective physical examination for confirming the diagnosis of lateral epicondylitis than the Cozen’s test, with consideration only for the different work activities, but not the physical activities performed by the patient.

According to a review, only a few studies have verified the reliability of Cozen’s and Mill’s tests, given their efficiency in provoking pain symptoms for the positive diagnosis of lateral epicondylitis, especially when the patient’s physical effort to practice physical activity is considered [18]. Based on this gap in the knowledge, the present study aimed to verify the reliability between these clinical tests according to the physical efforts directed to different levels of physical activity. Our results showed that the tests did not present good agreement accuracy with each other in the diagnosis of lateral epicondylitis. The accuracy of clinical tests is the basis for a good diagnosis of lateral epicondylitis; the tests are essential in reproducing the pain experienced by the patient [34]. We observed that regardless of the physical effort performed by the practice of PA, the Cozen’s and Mill’s tests lacked inter-test precision agreement, although Mill’s test was more effective in provoking the pain reported by the patient. Furthermore, the literature has highlighted the great need to verify the accuracy of clinical tests in relation to the imaging test considered for the diagnosis of lateral epicondylitis [14]. Among the imaging exams, ultrasonography has been shown to have a high sensitivity in the diagnosis of a pathological muscle-tendon disorder, exhibiting the presence of hypoechogenic fluid, tendon lacerations, micro-tears and echogenicity, as well as possible tendon calcifications [15,22,23,24]. In the current study, Mill’s test displayed excellent validity for the different levels of PA, supporting the ultrasound image examination in the positive diagnosis of lateral epicondylitis. Although Cozen’s test presented good accuracy with the ultrasound, it also showed false positive diagnoses in relation to the ultrasound exam, for all levels of physical activity evaluated.

Another important finding of this study was that the pain intensity caused by Cozen’s and Mill’s clinical tests was not related to the level of energy expenditure (MET/min/s) presented by the different levels of PA practice. We showed that the associated pain, caused by an inflammatory response in the tendon with lateral elbow epicondylitis, is not influenced by the level of physical activity of the affected individual. Thus, it can be inferred that the level of physical performance of the patient affected by epicondylitis did not influence the improvement or worsening of the clinical framework, as shown by the clinical tests for diagnosing the pathology. Haahr and Andersen [13] detected no association between lateral epicondylitis and physical activities outside work. In another study, Garg et al. [25] detected a significant association between swimming and lateral epicondylitis. It is worth mentioning that in the present study, we considered only individuals who already performed work activities related to physical effort (repetitive movements of the elbow and wrist/hand), with the practice of physical activity being considered as an activity outside of work.

In the literature, it is not yet clear whether the increase in physical stress caused by PA has consequences for patients with lateral epicondylitis who already have a mechanical overload promoted by work activities. According to Fan et al. [10], few studies have considered physical activities and hobbies outside of work as risk factors that contribute to lateral epicondylitis in workers. The majority of studies associate lateral epicondylitis only with physical efforts related to repetitive movements resulting from work activities [12,13,37,38]. The objective of this study was to show that in relation to the energy expenditure, the levels of physical activity did not promote changes in the clinical framework of pain in affected patients. Many studies have described the relationship between increased physical activity and a better inflammatory response compared to sedentary patients [39,40,41,42]. Although it was not the purpose of the study to evaluate the inflammatory markers of patients affected by lateral epicondylitis, it can be observed that although PA levels did not show any influence on the patient’s clinical symptoms, they presented an influence on the physical inspection of the patient. For example, Mill’s test proved to be more precise and accurate at different levels of physical activity (low, moderate and high) for the diagnosis of lateral epicondylitis, especially against ultrasound imaging.

A systematic review performed in 2021 following PRISMA-DTA guidelines showed that Cozen’s test presents high accuracy in the diagnosis of lateral epicondylitis, but has been poorly investigated. USI and MRI provide variable diagnostic accuracy depending on the entities reported and should be recommended with caution when a differential diagnosis is necessary. Substantial heterogeneity was detected in inclusion criteria, operator/examiner, mode of application, type of equipment and reference standards across the studies [43]. In the current study, we were careful to standardize the ultrasound exam, which was always performed by the same operator/examiner, mode of application, and type of equipment, and the results showed greater accuracy in relation to ultrasound with the Mill’s test in relation to the test from Cozen’s test, agreeing with Saroja et al. [22].

According to Ikeda et al., in 2022, individual evaluation of the common extensor tendon improved the severity diagnostic accuracy of the severity of lateral epicondylitis [44]. This fact shows the importance of the current study when considering Cozen’s and Mill’s clinical tests on the level of physical activity practice in the diagnosis of adults affected by lateral epicondylitis.

The limitation of this study was that it did not consider an experimental evaluation of inflammatory markers of the tendon associated with the practice of PA to better understand Cozen’s and Mill’s clinical tests in the diagnosis of lateral epicondylitis. Another important point is that it was not possible to control sports activities involving only the upper limbs, but we were careful not to consider soccer. Thus, future studies with this assessment approach are suggested to confirm the effectiveness of the clinical tests considered in this study.

## 5. Conclusions

Physical activity level influenced the clinical diagnostic tests (Cozen’s and Mill’s) of lateral elbow epicondylitis, in which the Mill’s test was more provocative of tendon pain. The inter-diagnostic test reliability was excellent for adults with a high level of physical activity. Mill’s test showed excellent accuracy for the different levels of physical activity (low, moderate and high) considered by ultrasound examination of the musculoskeletal tissue. The pain symptom caused by the Cozen’s or Mill’s test was not related to the energy expenditure (MET/min/s) obtained by different levels of physical activity practice.

## Figures and Tables

**Table 1 ijerph-20-01794-t001:** Mean, standard deviation and comparisons of the anthropometric and clinical variables among different groups of physical activity levels (PA): low, moderate and high in adults diagnosed with lateral epicondylitis.

Variables (Anthropometric and Clinical)	Low PA (*n* = 19)	Moderate PA (*n* = 42)	High PA (*n* = 41)	*p*
Age (years)	47.1 ± 7.3	45.1 ± 8.6	45.9 ± 8.1	0.779
Height (m)	1.6 ± 0.6	1.6 ± 0.8	1.6 ± 0.9	0.530
Body mass (kg)	71.6 ± 14.8	74.7 ± 17.6	75.1 ± 12.5	0.123
Body Mass Index (kg/m^2^)	22.1 ± 4.0	22.9 ± 5.1	22.6 ± 3.5	0.739
Gender (F/M)	15 (F) 4 (M)	30 (F) 12 (M)	24 (F) 17 (M)	0.286
Disease time (years)	7.3 ± 4.9	8.0 ± 5.3	9.4 ± 5.6	0.188
PA practice time (months)	20.4 ± 9.7	25.5 ± 9.8	41.0 ± 9.3	0.010 *

Legend: physical activity levels (PA). * One-way ANOVA test, considering statistical differences at *p* < 0.05.

**Table 2 ijerph-20-01794-t002:** Mean, standard deviation and comparisons of clinical inter-tests of pain symptoms and physical activity levels (PA): low, moderate, and high in adults’ elbow diagnosed with lateral epicondylitis.

Clinical Tests	Low PA	Moderate PA	High PA
Cozen (cm)	5.9 ± 2.5	6.5 ± 2.2	6.3 ± 2.6
Mill (cm)	8.2 ± 1.5	8.3 ± 1.4	7.7 ± 1.8
*p*	0.002 *	<0.001	<0.001

Legend: physical activity levels (PA). * Paired Student *t*-test, considering statistical differences at *p* < 0.05.

**Table 3 ijerph-20-01794-t003:** Inter-clinical reliability based on the intra-class agreement (ICC) index, standard error of measurement, and confidence interval for the different levels of physical activity (PA): low, moderate, and high in adults with lateral epicondylitis.

Physical Activity Level (PA)	Cozen (cm)	Mill (cm)	ICC	SEM	IC 95%	*p*
Low PA	5.9 ± 2.5	8.2 ± 1.5	0.14	0.64	0.12/0.17	0.037
Moderate PA	6.5 ± 2.2	8.3 ± 1.4	0.48	0.38	0.22/0.58	<0.001 *
High PA	6.3 ± 2.6	7.7 ± 1.8	0.44	0.44	0.39/0.69	0.024

Legend: physical activity levels (PA); Intra-Class Correlation Coefficient (ICC); Standard Error of Measurement (SEM); confidence interval—95% CI. * One-way ANOVA test, considering statistical differences at *p* < 0.05.

**Table 4 ijerph-20-01794-t004:** Agreement index of clinical tests with “gold standard” ultrasound examination in confirming the diagnosis of lateral epicondylitis in adults.

Ultrasound Examination Lateral Epicondylitis	Physical Activity Level (PA)	Cozen’s Test (*n*/%)	Mill’s Test (*n*/%)
Positive ultrasound	Low	15/78%	19/100%
Negative ultrasound		04/21%	0.0/0.0%
Kappa *		0.80	1.0
*p* value		0.042	0.997
Bland–Altman *		0.17	0.0
Positive ultrasound	Moderate	36/85.7%	38/90,4%
Negative ultrasound		06/14.2%	04/21%
Kappa *		0.74	0.82
*p* value		0.037	0.235
Bland–Altman *		0.14	0.11
Positive ultrasound	High	35/85.3%	40/97.5%
Negative ultrasound		06/14.6%	1/0.0%
Kappa *		0.73	0.99
*p* value		0.012	0.323
Bland–Altman *		0.14	0.02

* Kappa test and Bland–Altman analysis, considering statistical differences at *p* < 0.05.

**Table 5 ijerph-20-01794-t005:** Multiple linear regression analysis of the relationship between the energy expenditure (MET/s) of each physical activity level with the pain symptom caused during each clinical test (Cozen’s and Mill’s) in the diagnosis of lateral elbow epicondylitis in adults.

Physical Activity Level (PA)	Pain Cozen (cm)	R	R^2^	T	*p **
Low PA (MET/s)	5.9 ± 2.5	0.32	0.10	−0.95	0.354
Moderate PA (MET/s)	6.5 ± 2.2	0.18	0.03	1.12	0.268
High PA (MET/s)	6.3 ± 2.6	0.21	0.04	−0.05	0.962
	**Pain** **Mill (cm)**	**R**	**R^2^**	**T**	** *p* **
Low PA (MET/s)	8.2 ± 1.5	0.31	0.10	−0.92	0.371
Moderate PA (MET/s)	8.3 ± 1.4	0.02	0.01	−0.07	0.932
High PA (MET/s)	7.7 ± 1.8	0.26	0.07	−0.89	0.376

* Multiple Linear Regression Analysis Model, considering statistical differences at *p* < 0.05.

## Data Availability

The datasets generated and/or analyzed during the current study are not publicly available due to limitations of ethical approval involving the patient data and anonymity but are available from the corresponding author (apribeiro@alumni.usp.br) upon reasonable request.

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
