# Peer review of "Differences in Clinical Tests for Assessing Lateral Epicondylitis Elbow in Adults Concerning Their Physical Activity Level: Test Reliability, Accuracy of Ultrasound Imaging, and Relationship with Energy Expenditure"

_ijerph, 2023, doi:10.3390/ijerph20031794_

Round 1
Reviewer 1 Report
This article has the following major drawbacks reviewing the first two sections (Introduction and Methods):
- The introduced aim of the study in the Introduction is not very clear since it contains several objectives and I suggest introducing it through several hypotheses. In addition, it is to be directly reflected in the Abstract.
- The sampling process is not clear. I think the sample size is not enough to get valid results and in this regard, the sample size process is to be presented with details.
Addressing the mentioned drawbacks may change the structure of the article substantially. Thus, I prefer to continue reviewing the other sections after evaluating how the authors would address these issues.
Author Response
São Paulo, 03th of December 2022
International Journal of Environmental Research and Public Health - IJERPH
Dear Editor-in-Chief,
Prof. Dr. Javier Abián-Vicén, Prof. Dr. Prof. Dr. José Carmelo Adsuar Sala and Prof. Dr. Stefano Ballestri
Editor and Editorial Board Member of International Journal of Environmental Research and Public Health,
We, the authors, would like to resubmit the paper “Physical Activity Level on Clinical Tests for Assessing Lateral Epicondylitis Elbow in Adults: Reliability, Accuracy by Ultrasound Imaging, and Relations with Energy Expenditure” (ijerph-2070751) in a revised form, as suggested. We are sending also a covering letter responding to the reviewer’s comments on a point-by-point basis. Bellow, we followed your comments and answered each one also in a point-by-point basis. The authors would like to thank the editor for the careful revision and constructive comments/suggestions on our manuscript that certainly contributed for a better version of it.
Yours sincerely,
The authors
Responses to the reviewers' comments on a point-by-point basis
Submission: Ref: Submission (ijerph-2070751)
Reviewer #1:
We thank the reviewer for the constructive comments on our manuscript. Your suggestions and remarks have helped us to reflect on the manuscript and make a better version. We appreciate your suggestions and carefully considered every one of your comments and we made the appropriate changes. Below, we responded to your remarks on a point-by-point basis and inserted the corrections to the new version of the manuscript (underlined parts). Some of your comments will be discussed here. First your comment is given in bold; subsequently we provide our answer.
Comments and Suggestions for Authors
- This article has the following major drawbacks reviewing the first two sections (Introduction and Methods): - The introduced aim of the study in the Introduction is not very clear since it contains several objectives and I suggest introducing it through several hypotheses. In addition, it is to be directly reflected in the Abstract.
Answer: We appreciate your comment and agree with the reviewer. In this way, we synthesized the objective to make it more and more specific in just one purpose in the introduction and abstract. In addition, we add the study hypotheses, after the objective, in the introduction section. Page 1 (line 14-16); Page 3 (line 113-119). Underlined parts.
- The sampling process is not clear. I think the sample size is not enough to get valid results and in this regard, the sample size process is to be presented with details.
Answer: We very appreciate your comment and all your attention in helping us to improve the manuscript. We detail and further explain the sampling calculation in the statistics section. We strongly hope that we have heeded the reviewer's comment. Page 5 (line 209-214). Underlined parts.
- Addressing the mentioned drawbacks may change the structure of the article substantially. Thus, I prefer to continue reviewing the other sections after evaluating how the authors would address these issues.
Answer: We hope that we addressed the reviewer's comments and hope that you will continue to review our article. Thank you for everything.

Reviewer 2 Report
Dear Sir/ Madam,
The topic of the paper is very interesting and it was my pleasure to read it.
Suggestion for the title is “Differences in results of Clinical Tests for Assessing Lateral Epicondylitis Elbow in Adults concerning their Physical Activity Level: Tests Reliability, Accuracy by Ultrasound Imaging, and Relations with Energy Expenditure"
The methodology of the research fits the studied subject. Professional terminology is properly used. As a reviewer of this paper, I find that there are no errors in theoretical presentation, but there are some suggestions to correct in the paper:
1. Each table should have legend explaining the abbreviations used in the table
2. In Table 1 words “Baixa AF, Moderada AF and Alta AF” are not translated to English, also Disease time and PA practise time are not anthropometric variables there are clinical variables and that should be in table two.
3. Table 2 is not paired student t-test and table 2 is the same as table 3, i suggest correction of table 2.
4. Page 6 line 233 capital letter for Kappa
5. Page 7 line265 instead of mild use word low (PA levels)
6. Page 7 line 277 “do not differ between different groups of patients” with different “levels of physical activity”,
7. Page 8 line 337 instead of mild use word low (PA levels),
Parts of the paper are of the appropriate extent and there are no unnecessary repetitions in the text. The text is written clearly and logically, and the conclusion is drowning from the results obtained. As for the literature, references are written correctly.
Best regards
Author Response
São Paulo, 03th of December 2022
International Journal of Environmental Research and Public Health - IJERPH
Dear Editor-in-Chief,
Prof. Dr. Javier Abián-Vicén, Prof. Dr. Prof. Dr. José Carmelo Adsuar Sala and Prof. Dr. Stefano Ballestri
Editor and Editorial Board Member of International Journal of Environmental Research and Public Health,
We, the authors, would like to resubmit the paper “Physical Activity Level on Clinical Tests for Assessing Lateral Epicondylitis Elbow in Adults: Reliability, Accuracy by Ultrasound Imaging, and Relations with Energy Expenditure” (ijerph-2070751) in a revised form, as suggested. We are sending also a covering letter responding to the reviewer’s comments on a point-by-point basis. Bellow, we followed your comments and answered each one also in a point-by-point basis. The authors would like to thank the editor for the careful revision and constructive comments/suggestions on our manuscript that certainly contributed for a better version of it.
Yours sincerely,
The authors
Responses to the reviewers' comments on a point-by-point basis
Submission: Ref: Submission (ijerph-2070751)
Reviewer #2:
We thank the reviewer for the constructive comments on our manuscript. Your suggestions and remarks have helped us to reflect on the manuscript and make a better version. We appreciate your suggestions and carefully considered every one of your comments and we made the appropriate changes. Below, we responded to your remarks on a point-by-point basis and inserted the corrections to the new version of the manuscript (underlined parts). Some of your comments will be discussed here. First your comment is given in bold; subsequently we provide our answer.
Reviewer's report
English language and style are fine/minor spell check required.
Answer: We appreciated your comment and strongly apologize for the English language in the manuscript. We have submitted the manuscript to a professional service for reviewing English as a second language (Scribendi). We hope it achieves the standards of this respectful Journal (Journal of Clinical Medicine).
Comments and Suggestions for Authors
- Dear Sir/ Madam,
The topic of the paper is very interesting and it was my pleasure to read it.
Suggestion for the title is “Differences in results of Clinical Tests for Assessing Lateral Epicondylitis Elbow in Adults concerning their Physical Activity Level: Tests Reliability, Accuracy by Ultrasound Imaging, and Relations with Energy Expenditure"
Answer: We thank the reviewer for work and help in the best manuscript. We appreciate all the comments and suggestions. We accept and appreciate your suggestion on the title and change it as proposed. Page 1 (line 2-5). Underlined parts.
- The methodology of the research fits the studied subject. Professional terminology is properly used. As a reviewer of this paper, I find that there are no errors in theoretical presentation, but there are some suggestions to correct in the paper:
- Each table should have legend explaining the abbreviations used in the table
Answer: We very appreciate your comment and we have added this information of the legend in each table. Page 5 (Table 1, line 242); Page 6 (Table 2 line 253; Table 3 line 263-264); Page 7 (Table 4 line 283; Table 5 line 302-303). Underlined parts.
- In Table 1 words “Baixa AF, Moderada AF and Alta AF” are not translated to English, also Disease time and PA practise time are not anthropometric variables there are clinical variables and that should be in table two.
Answer: We appreciated your comment and strongly apologize for the error in the translation into English language. We corrected this in the text and specified the variables in the table, as suggested by the reviewer. In this way, we added this information from the sports activities practiced by the participants and full sample characteristics in the methods session. Page 3 (line 123-133); Page 5 (Table 1 line 239-242). Underlined parts.
- Table 2 is not paired student t-test and table 2 is the same as table 3, i suggest correction of table 2.
Answer: We really apologize for this error, and we have corrected table 2. Thank you very much for your help and excellent work. Page 6 (Table 2 line 251-253). Underlined parts.
- Page 6 line 233 capital letter for Kappa.
Answer: We appreciate and thank the reviewer very much and this error has been corrected in the text. Page 6 (line 267). Underlined parts.
- Page 7 line265 instead of mild use word low (PA levels).
Answer: We appreciate and thank the reviewer very much and this error has been corrected in the text. Page 8 (line 312). Underlined parts.
- Page 7 line 277 “do not differ between different groups of patients” with different“levels of physical activity”.
Answer: We appreciate and thank the reviewer very much and this error has been corrected in the text. Page 8 (line 324). Underlined parts.
- Page 8 line 337 instead of mild use word low (PA levels).
Answer: We appreciate and thank the reviewer very much and this error has been corrected in the text. Page 9 (line 384). Underlined parts.
- Parts of the paper are of the appropriate extent and there are no unnecessary repetitions in the text. The text is written clearly and logically, and the conclusion is drowning from the results obtained. As for the literature, references are written correctly.
Answer: I have no words to thank you for your comments, which, without a doubt, helped us to make the information in the manuscript clearer, more coherent and with an improvement in scientific writing based on the literature. We would like to reiterate our thanks for your work, efficiency and attention in helping us. We strongly hope that we have taken note of your comments in detail, and that they are in line with your appreciation.

Reviewer 3 Report
Dear Authors,
I read your work with interest. The reviewed manuscript is logically structured in parts, the abstract summarizes the essential aspects of the research, the directions of the research are mentioned at the end of the introduction, the results are scientifically well argued. I appreciate the quality of the reviewed article and can only provide you with a few minor ideas/suggestions to improve the current version:
1. You stated in the abstract that for the clinical tests (Cozen and Mill tests) you have used the VAS (Visual Analogue Scale) with a graduated ruler (cm) to assess the pain intensity. Perhaps it would be useful to specify this idea and the interpretation of the scale in Evaluation of clinical test (line134-150).
2. Table 1 would be more useful in the Participants section, not in Results. Table 1 (top): PA levels are not in English (Baixa AF, Moderada AF, Alta AF).
3. Tables 2 and 3 are identical (variables and data presented), their headings indicate different aspects.
4. Could you specify the average age of the investigated group, the number of women and men, if significant gender differences are identified for the investigated variables?
5. In the Introduction (lines 62-65) you mentioned the professional categories and sports that generate pain in the elbow joint (through overuse and inflammation of the extensor carpi radialis brevis/ECRB). What are the professions with the highest frequency of Lateral Epicondylitis (Tennis elbow) in the group you investigated?
6. You divided the participants into 3 physical activity levels (PA). Lines 318-321 indicate that all selected participants have professions that involve strong demands on the elbow joint. Lines 329-331: The differential of this study was to show that the levels of physical activity, in relation to the energy expenditure, did not promote changes in the clinical pain picture of the affected patients. However, the physical sports activities of free (leisure) time can have different specific articular demands: (I think that Jogging and Soccer players do not have the same demands on the elbow joint as tennis players, different types of martial arts practitioners, etc. do). In the first case, a large volume of physical activity and very high energy consumption can be achieved, but without excessive/repetitive strain on the elbow and hand extensors. Was this controlled for in your study?
I congratulate you on the scientific article and wish you success in your research activities.
Author Response
São Paulo, 03th of December 2022
International Journal of Environmental Research and Public Health - IJERPH
Dear Editor-in-Chief,
Prof. Dr. Javier Abián-Vicén, Prof. Dr. Prof. Dr. José Carmelo Adsuar Sala and Prof. Dr. Stefano Ballestri
Editor and Editorial Board Member of International Journal of Environmental Research and Public Health,
We, the authors, would like to resubmit the paper “Physical Activity Level on Clinical Tests for Assessing Lateral Epicondylitis Elbow in Adults: Reliability, Accuracy by Ultrasound Imaging, and Relations with Energy Expenditure” (ijerph-2070751) in a revised form, as suggested. We are sending also a covering letter responding to the reviewer’s comments on a point-by-point basis. Bellow, we followed your comments and answered each one also in a point-by-point basis. The authors would like to thank the editor for the careful revision and constructive comments/suggestions on our manuscript that certainly contributed for a better version of it.
Yours sincerely,
The authors
Responses to the reviewers' comments on a point-by-point basis
Submission: Ref: Submission (ijerph-2070751)
Reviewer #3:
We thank the reviewer for the constructive comments on our manuscript. Your suggestions and remarks have helped us to reflect on the manuscript and make a better version. We appreciate your suggestions and carefully considered every one of your comments and we made the appropriate changes. Below, we responded to your remarks on a point-by-point basis and inserted the corrections to the new version of the manuscript (underlined parts). Some of your comments will be discussed here. First your comment is given in bold; subsequently we provide our answer.
Reviewer's report
Comments and Suggestions for Authors
- Dear Authors,
I read your work with interest. The reviewed manuscript is logically structured in parts, the abstract summarizes the essential aspects of the research, the directions of the research are mentioned at the end of the introduction, the results are scientifically well argued. I appreciate the quality of the reviewed article and can only provide you with a few minor ideas/suggestions to improve the current version:
Answer: We greatly appreciate your comments, help, and all the work you do to help us improve the manuscript. Thank you very much for your excellent work.
- You stated in the abstract that for the clinical tests (Cozen and Mill tests) you have used the VAS (Visual Analogue Scale) with a graduated ruler (cm) to assess the pain intensity. Perhaps it would be useful to specify this idea and the interpretation of the scale in Evaluation of clinical test (line134-150).
Answer: We very appreciate your comment and we really thank your observation. We have added this information in session methods. In addition, the reference corresponding reference has been added. Page 4 (line 160-162); Page 10 (line 465-467). Underlined parts.
- Johnson C. Measuring pain—visual analog scale versus numeric pain scale: What is the difference? J Chiropr Med. 2005;4(1):43–4.
- Table 1 would be more useful in the Participants section, not in Results. Table 1 (top): PA levels are not in English (Baixa AF, Moderada AF, Alta AF).
Answer: We appreciated your comment and strongly apologize for the error in the translation into English language. However, we left the description of the anthropometric and professional characteristics of the total sample in the methods section, but we had to keep Table 1, at the request of reviewer 2, with the specific characteristics in each group evaluated. We hope that the reviewer can understand, if he still thinks it is necessary, we change it and separate it from the clinical variables. Many thanks, again. Page 5 (Table 1 line 241-242). Underlined parts.
- Tables 2 and 3 are identical (variables and data presented), their headings indicate different aspects.
Answer: We really apologize for this error, and we have corrected table 2. Thank you very much for your help and excellent work. Page 6 (Table 2 line 251-253). Underlined parts.
- Could you specify the average age of the investigated group, the number of women and men, if significant gender differences are identified for the investigated variables?
Answer: We appreciate your comments and would like to point out that we have added the full sample characteristics in the methods section and also added the age and gender specificity in table 1 according to the specificity of the physical activity level group. We hope to have taken into account your comments. Page 3 (line 126-132); Page 6 (Table 1, line 241-242). Underlined parts.
- In the Introduction (lines 62-65) you mentioned the professional categories and sports that generate pain in the elbow joint (through overuse and inflammation of the extensor carpi radialis brevis/ECRB). What are the professions with the highest frequency of Lateral Epicondylitis (Tennis elbow) in the group you investigated?
Answer: We appreciate and thank the reviewer very much. We added the work characteristics (professional) and sports activities of the participants in the methods section. Page 3 (line 126-133). Underlined parts.
- You divided the participants into 3 physical activity levels (PA). Lines 318-321 indicate that all selected participants have professions that involve strong demands on the elbow joint. Lines 329-331: The differential of this study was to show that the levels of physical activity, in relation to the energy expenditure, did not promote changes in the clinical pain picture of the affected patients. However, the physical sports activities of free (leisure) time can have different specific articular demands: (I think that Jogging and Soccer players do not have the same demands on the elbow joint as tennis players, different types of martial arts practitioners, etc. do). In the first case, a large volume of physical activity and very high energy consumption can be achieved, but without excessive/repetitive strain on the elbow and hand extensors. Was this controlled for in your study?
Answer: We really appreciate and agree with your excellent comment. However, we were unable to control the sports activities carried out only by the upper limbs, but we were careful not to consider soccer. In this way, we added this information from the sports activities practiced by the participants in the methods session. Furthermore, we added this question as a study limitation in the discussion session. Page 3 (line 131-133). Underlined parts.
- I congratulate you on the scientific article and wish you success in your research activities.
Answer: I have no words to thank you for your comments, which, without a doubt, helped us to make the information in the manuscript clearer, more coherent and with an improvement in scientific writing based on the literature. We would like to reiterate our thanks for your work, efficiency and attention in helping us. We strongly hope that we have taken note of your comments in detail, and that they are in line with your appreciation.

Round 2
Reviewer 1 Report
This study has the following major drawbacks:
- The presented aim and the following presented hypotheses are still complicated. I suggested using the hypotheses to be able to clarify different aspects of your study. But the presented hypothesis does not help in this respect. For instance, the presented hypotheses could be divided into various separate hypotheses. In the case of using such separated hypotheses, they could be also included in the section of Results.
- The criterion of dividing the participants into three levels of physical activity is to be directly mentioned.
- A better clarification is required to explain the used regression analysis concerning energy expenditure and the different types of variables used in this analysis.
- Few studies have been cited in the section of Discussion. I think this section could be enriched with more studies from different contexts.
- The section on Conclusions is to be extended by a brief explanation of the aim and objectives of the study.
Author Response
São Paulo, 14th of December 2022
International Journal of Environmental Research and Public Health - IJERPH
Dear Editor-in-Chief,
Prof. Dr. Javier Abián-Vicén, Prof. Dr. Prof. Dr. José Carmelo Adsuar Sala and Prof. Dr. Stefano Ballestri
Editor and Editorial Board Member of International Journal of Environmental Research and Public Health,
We, the authors, would like to resubmit the paper “Physical Activity Level on Clinical Tests for Assessing Lateral Epicondylitis Elbow in Adults: Reliability, Accuracy by Ultrasound Imaging, and Relations with Energy Expenditure” (ijerph-2070751) in a revised form, as suggested. We are sending also a covering letter responding to the reviewer’s comments on a point-by-point basis. Bellow, we followed your comments and answered each one also in a point-by-point basis. The authors would like to thank the editor for the careful revision and constructive comments/suggestions on our manuscript that certainly contributed for a better version of it.
Yours sincerely,
The authors
Responses to the reviewers' comments on a point-by-point basis
Submission: Ref: Submission (ijerph-2070751)
Reviewer #1 (2 revision):
We thank the reviewer for the constructive comments on our manuscript. Your suggestions and remarks have helped us to reflect on the manuscript and make a better version. We appreciate your suggestions and carefully considered every one of your comments and we made the appropriate changes. Below, we responded to your remarks on a point-by-point basis and inserted the corrections to the new version of the manuscript (underlined parts). Some of your comments will be discussed here. First your comment is given in bold; subsequently we provide our answer.
Comments and Suggestions for Authors
- This study has the following major drawbacks:
- The presented aim and the following presented hypotheses are still complicated. I suggested using the hypotheses to be able to clarify different aspects of your study. But the presented hypothesis does not help in this respect. For instance, the presented hypotheses could be divided into various separate hypotheses. In the case of using such separated hypotheses, they could be also included in the section of Results.
Answer: We appreciate your comment. We have structured the objectives to better specify, as well as detailing the hypotheses. Page 3 (line 113-125). Underlined parts.
- The criterion of dividing the participants into three levels of physical activity is to be directly mentioned.
Answer: We appreciate your comment. However, we had already detailed the entire process of dividing the groups by physical activity level in the methods section, however, we left it in more detail for a better understanding. Page 4 (line 150-167). Underlined parts.
- A better clarification is required to explain the used regression analysis concerning energy expenditure and the different types of variables used in this analysis.
Answer: We appreciate your comment. Simple linear regression is a mathematical technique used to model the relationship between a single independent predictor variable and a single dependent outcome variable. Thus, this study the simples linear regression analyses were used to predict the dependent variables (pain provoked by the clinical diagnostic tests: Cozen and Mil) and the level of physical activity. The dependent variable (pain Cozen and pain Mill) was included in the model by correlation coefficients were higher than 0.30 were entered into the model. For all of the analyses, we adopted p<0.05. Page 5 (line 235-242). Underlined parts.
- Few studies have been cited in the section of Discussion. I think this section could be enriched with more studies from different contexts.
Answer: We appreciate and consider your comment. However, in total of 20 references were directed to the discussion, given the total of 43 references. To respond to your comment, we considered adding two current references, corresponding to the years 2021 and 2022, to expand and discuss more specifically the observed results, as shown below. Page 9 (line 403-405); Page 10 (line 406-417); Page 12 (line 531-536). Underlined parts.
- 44. Karanasios S, Korakakis V, Moutzouri M, Drakonaki E, Koci K, Pantazopoulou V, Tsepis E, Gioftsos G. Diagnostic ac-curacy of examination tests for lateral elbow tendinopathy (LET) - A systematic review. J Hand Ther. 2021 Feb 27:S0894-1130(21)00039-9. doi: 10.1016/j.jht.2021.02.002.\
- 45. Ikeda K, Ogawa T, Ikumi A, Yoshii Y, Kohyama S, Ikeda R, Yamazaki M. Individual Evaluation of the Common Exten-sor Tendon and Lateral Collateral Ligament Improves the Severity Diagnostic Accuracy of Magnetic Resonance Imag-ing for Lateral Epicondylitis. Diagnostics (Basel). 2022 Aug 2;12(8):1871. doi: 10.3390/diagnostics12081871.
- The section on Conclusions is to be extended by a brief explanation of the aim and objectives of the study.
Answer: We appreciate your comment. We rewrote the entire conclusion based on responding to our objectives and hypotheses. Page 10 (line 426-432). Underlined parts.
